

# Bad metals from fluctuating density waves

**Luca V. Delacrétaz[1], Blaise Goutéraux[1,2,3], Sean A. Hartnoll[1*] and Anna Karlsson[1]**

**1** Department of Physics, Stanford University,
Stanford, CA 94305-4060, USA
**2** APC, Université Paris 7, CNRS, CEA,
Observatoire de Paris, Sorbonne Paris Cité, F-75205, Paris Cedex 13, France
**3** Nordita, KTH Royal Institute of Technology and Stockholm University,
Roslagstullsbacken 23, SE-106 91 Stockholm, Sweden

⋆ hartnoll@stanford.edu

## Abstract

**Bad metals have a large resistivity without being strongly disordered. In many bad metals the Drude peak moves away from zero frequency as the resistivity becomes large at increasing temperatures. We catalogue the position and width of the 'displaced Drude peak' in the observed optical conductivity of several families of bad metals, showing that $\omega_{\text{peak}} \sim \Delta\omega \sim k_B T/\hbar$. This is the same quantum critical timescale that underpins the $T$-linear dc resistivity of many of these materials. We provide a unified theoretical description of the optical and dc transport properties of bad metals in terms of the hydrodynamics of short range quantum critical fluctuations of incommensurate density wave order. Within hydrodynamics, pinned translational order is essential to obtain the nonzero frequency peak.**



Bad metals are defined by the fact that if their electrical resistivity is interpreted within a conventional Drude formalism, the corresponding mean free path of the quasiparticles is so short that the Boltzmann equation underlying Drude theory is not consistent [1–3]. As such, bad metals pose a long-standing challenge to theory. In this work we show that the hydrodynamics of phase-fluctuating charge density waves can lead to bad metallic behavior. Hydrodynamics is a tightly constrained formalism for the low energy and long wavelength dynamics of media [4, 5]. Non-quasiparticle media, in particular, exhibit fast local thermalization leading to extended hydrodynamics regimes. Phase fluctuations in the density wave can be robustly incorporated into this description and will be essential in order for the states to be metallic, rather than insulating. We will use the hydrodynamic framework to explain observed dc and optical transport behavior that is common to several families of bad metal materials.

Recent work has emphasized that the absence of quasiparticles is not sufficient to obtain a bad metal [6]. If the total momentum of the charge carriers is long-lived, then the resistivity will be small even if all single-particle excitations decay rapidly. The importance of the fate of momentum for transport has long been appreciated [7,8], but has acquired renewed relevance

in the context of modern unconventional metals, e.g. [9, 10]. Two previously considered scenarios for removing the long-lived momentum (sound) mode from the collective description of charge transport are as follows. Firstly, that the low energy, non-quasiparticle, description of bad metals is strongly non-translationally invariant and hence momentum is absent from the hydrodynamic description [6]. Secondly, that an emergent particle-hole symmetry at low energies decouples charge transport from momentum [11]. These mechanisms do not seem to be at work in bad metals. The resistivity of bad metals is not typically strongly dependent on the strength of disorder and some bad metals appear to be relatively clean. Emergent particle-hole symmetry does not decouple momentum from heat transport and hence leads to a substantial violation of the Wiedemann-Franz law with Lorenz number $L \gg L_0$. Such behavior has recently been observed in clean graphene near the particle-hole symmetric point [12], but is not observed in bad metals.

In this paper we show that there is an alternative path to bad metallic behavior that allows for a large resistivity even in clean materials with a long-lived momentum. These will be non-quasiparticle states with weakly pinned, phase-fluctuating incommensurate density wave order. Our discussion will be in terms of charge density waves but spin density waves, or any other type of translational order, lead to similar physics. In a charge ordered state, the momentum zero mode becomes the Goldstone mode of the translational order. It is well known that weak disorder (or any form of explicit translation symmetry breaking) both gaps and broadens this mode, pinning the charge density wave [13–15]. The gap of this pseudo-Goldstone mode leads to fundamentally distinct low energy physics compared to the case without charge order, in which the zero mode is broadened by disorder into a Drude peak but not gapped. Due to this gap, the dc conductivity is disassociated from any weak momentum relaxation rate: the conductivity need not be large even in a weakly disordered limit. In fact, as is well known, if the underlying clean theory is Galilean invariant, then the dc conductivity is zero and the state is insulating [13–15]. A new aspect of our discussion will be that in non-Galilean invariant cases, or with phase fluctuations, the state can instead be metallic, with universal diffusive processes determining the dc conductivity.

A small dc conductivity despite only weak disorder is one of two key phenomenological consequences of the charge density wave scenario for bad metals. The second consequence is a peak in the optical conductivity away from $\omega = 0$. This is nothing other than the gapped pseudo-Goldstone mode or 'pinned phason' of the previous paragraph. It is in fact observed that in a large number of bad metals the Drude peak in the optical conductivity moves away from $\omega = 0$ as the temperature is increased, leading to a suppression of low frequency spectral weight. This phenomenon is seen in ruthenates [16,17], cobaltates [18,19], cuprates [20–24], vanadates [25,26], manganates [27,28], nickelates [29] and organic conductors [30,31]. The behavior is illustrated in figure 1 below.

We emphasize at the outset that we are *not* discussing the 'mid infrared bands' that are also observed at low temperatures in some of these materials [32]. We are interested specifically in circumstances where the Drude peak itself moves away from $\omega = 0$ as the temperature is *increased*. The 'mid infrared' peaks, in contrast, can coexist with well defined Drude components at low temperatures and furthermore typically occur at far higher frequencies, with $\hbar\omega \gg k_B T$, than the peaks we will be discussing [33]. The peaks are also not optical phonon peaks, these can occur in the same frequency range but are much sharper.

A variety of microscopic mechanisms are believed to cause the behavior shown in figure 1. Such charge dynamics is often discussed, for example, in terms of Mott-related pseudo-gaps, incipient localization or polaron excitations [32]. Indeed, many bad metals are close to localization transitions and/or show strong electron-phonon coupling. Given the strongly correlated, non-quasiparticle nature of the dynamics – as well as *a priori* surprising similarities in transport among distinct families of materials – we wish to attack the problem from a

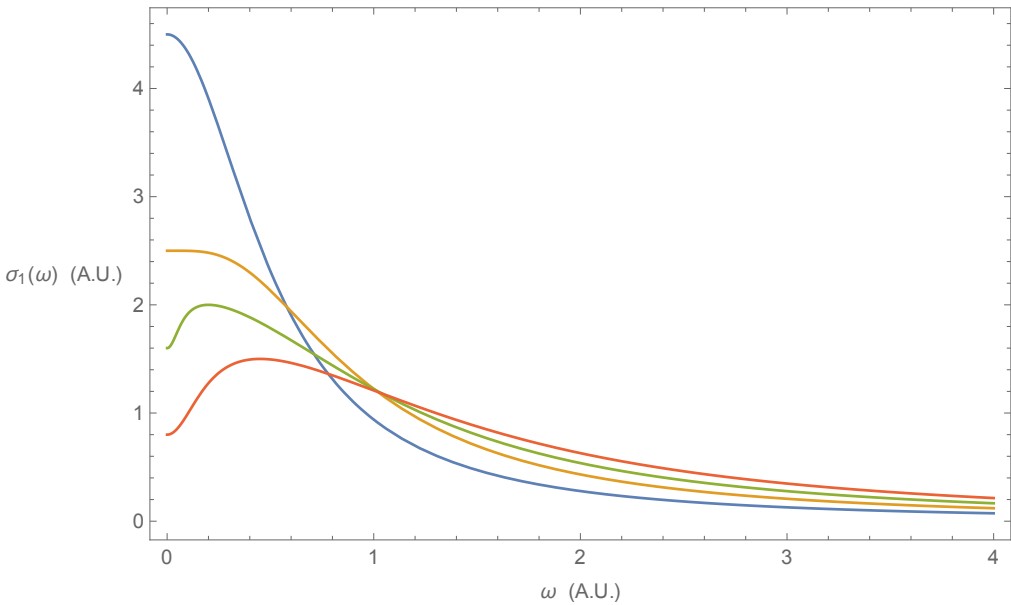

Figure 1: Illustrative plot of the temperature dependence of the optical conductivity of bad metals. As temperature is increased, the peak broadens and then moves off the $\omega = 0$ axis.

different, less microscopic, angle. Our objective is to determine the appropriate universality class of low energy dynamics leading to the behavior in figure 1, and in particular to identify the nature of any collective low energy observables. This allows a unified discussion of the different materials, in the spirit of [6, 34].

In non-quasiparticle systems, local thermalization is fast because the only long-lived excitations are associated with conserved or almost-conserved quantities. Hydrodynamics is the correct framework to describe the spatial and temporal relaxation of these long-lived modes. The crucial point, then, is that within hydrodynamics, it is not possible to obtain the dynamics of figure 1 from momentum relaxation alone. Momentum relaxation will broaden the Drude peak but does not gap the peak.[1] A hydrodynamic peak at nonzero frequencies, that is continuously connected to the Drude peak, is tantamount to saying that the momentum zero mode has become a pseudo-Goldstone boson. That is to say, a weakly pinned charge density wave. The fact that pinned charge density waves are the natural non-quasiparticle collective mode that leads to the response shown in figure 1 has previously been noted in [35]. That paper, as well as [36, 37], have furthermore shown that the nonzero frequency peak can be created and pushed to higher frequencies by increased disorder. This is the behavior expected for density waves due to increased pinning. This effect can also be seen by comparing the positions of the peak in [20] and [38], which are cleaner and dirtier versions of the same material.

The hydrodynamics of charge density waves is an old subject that is discussed in textbooks [4]. However, in order to describe the conductivity of bad metals it is crucial to include both the effects of broken translation invariance and of phase fluctuations. The consequences of broken translation invariance are well known, leading to insulating behavior due to the pinning of the density wave [13–15]. We will see that metallic behavior is obtained by incorporating phase fluctuations due to proliferating dislocations into the hydrodynamics. Dislocations are vortices in the phase of the density wave, and are therefore essential in order to dynamically relax phase gradients. The Galilean hydrodynamics of charge density wave states with mobile dislocations and defects was developed some time ago in [39]. In [40] we have worked out

---

[1]We are assuming the absence of magnetic fields or other sources of time-reversal symmetry breaking that can lead to cyclotron-type modes.

in detail the mathematical theory of charge density wave hydrodynamics that incorporates both momentum and phase-relaxing dynamics, and characterized the corresponding charge transport. In the supplementary material we outline the key steps in this derivation. The central result for the optical conductivity is

$$\sigma(\omega) = \sigma_o + \frac{\rho^2}{\chi_{\pi\pi}} \frac{\Omega - i\omega}{(\Omega - i\omega)(\Gamma - i\omega) + \omega_o^2} \,. \tag{1}$$

Here $\Gamma$ is the momentum relaxation rate, $\Omega$ is the phase relaxation rate (of the charge density wave), $\omega_o$ is the pinning frequency, $\rho^2/\chi_{\pi\pi}$ is the Drude weight and $\sigma_o$ is an 'incoherent' conductivity that can arise once Galilean invariance is broken. Hydrodynamics is a tightly constrained framework, and these are all the parameters that can appear within the long wavelength hydrodynamic description. In particular, the phase-relaxing physics of dislocations is uniquely captured by the parameter $\Omega$.

Putting $\Omega = \sigma_o = 0$ in (1) recovers the well-known expression for pinned charge density waves [15]. This is an insulator with $\sigma_{\mathrm{dc}} = 0$. Taking the $\Omega \to \infty$ limit of strong phase relaxation in (1) recovers the standard Drude expression, as we should expect. More generally, (1) describes a nontrivial interplay of phase and momentum relaxation.

The optical response (1) has two important properties. Firstly, if the pinning frequency is sufficiently large, $\omega_o^2 > \Omega^3/(\Gamma + 2\Omega)$, then the real part of the conductivity (1) has a peak away from $\omega = 0$. Secondly, taking $\omega \to 0$ gives the dc conductivity

$$\sigma_{\mathrm{dc}} = \sigma_o + \frac{\rho^2}{\chi_{\pi\pi}} \frac{\Omega}{\Omega\Gamma + \omega_o^2} \,. \tag{2}$$

The formula has a 'parallel resistor' form, typical of hydrodynamic regimes, in which two different effects, $\sigma_o$ and terms related to the charge density wave, add in the conductivity. The key feature of (2) is that the momentum relaxation rate $\Gamma$ can be taken to zero with the conductivity remaining finite and nonzero.

We are now ready to take a closer look at the data. Our focus will be on the materials mentioned above figure 1. In these materials, the optical conductivity $\sigma(\omega)$ peaks away from zero frequency at high temperatures, in the regime of bad metallic transport. Figure 2 shows the position and width of the peak in $\sigma(\omega)$ in units of temperature – $\hbar\omega_{\mathrm{peak}}/k_B T$ and $\hbar\Delta\omega/k_B T$ – plotted against the dc conductivity of the same material at the same temperature. We are interested in high temperature measurements of metals with large resistivity. It is a log-log plot. There is no fitting done: the peak location is the frequency at which the $\sigma(\omega)$ data has a maximum. The peak width is defined as the distance between the two points on either side of the maximum in $\sigma(\omega)$ which are at 90% of the height of the maximum. The immediate conclusion is that for all the materials and temperatures shown in this plot, the peak and width satisfy

$$\hbar\omega_{\mathrm{peak}} = \alpha(T) k_B T \,, \qquad \hbar\Delta\omega = \beta(T) k_B T \,, \tag{3}$$

with the coefficients $\alpha$ and $\beta$ material dependent but roughly in the range $0.2-4$. Some materials show small differences between the measured dc conductivity and the $\omega \to 0$ extrapolation of the optical data, which may be due to a very small Drude-like component at low frequencies. In some cases $\alpha$ and $\beta$ have a weak temperature dependence for a fixed material. Other materials show a scaling collapse in $\omega/T$. We show the temperature dependence of the peak locations in the supplementary material. That plot includes data on the manganates, omitted from the plot above because the large magnitude and stronger temperature dependence of the peak locations suggests different and likely non-hydrodynamic physics is at work.

The 'Planckian' [42] timescale $\tau \sim \hbar/(k_B T)$ revealed in figure 2 and equation (3) is characteristic of quantum critical systems [43]. Quantum criticality describes the competition between order and quantum fluctuations. The footprint of a quantum critical point on the phase

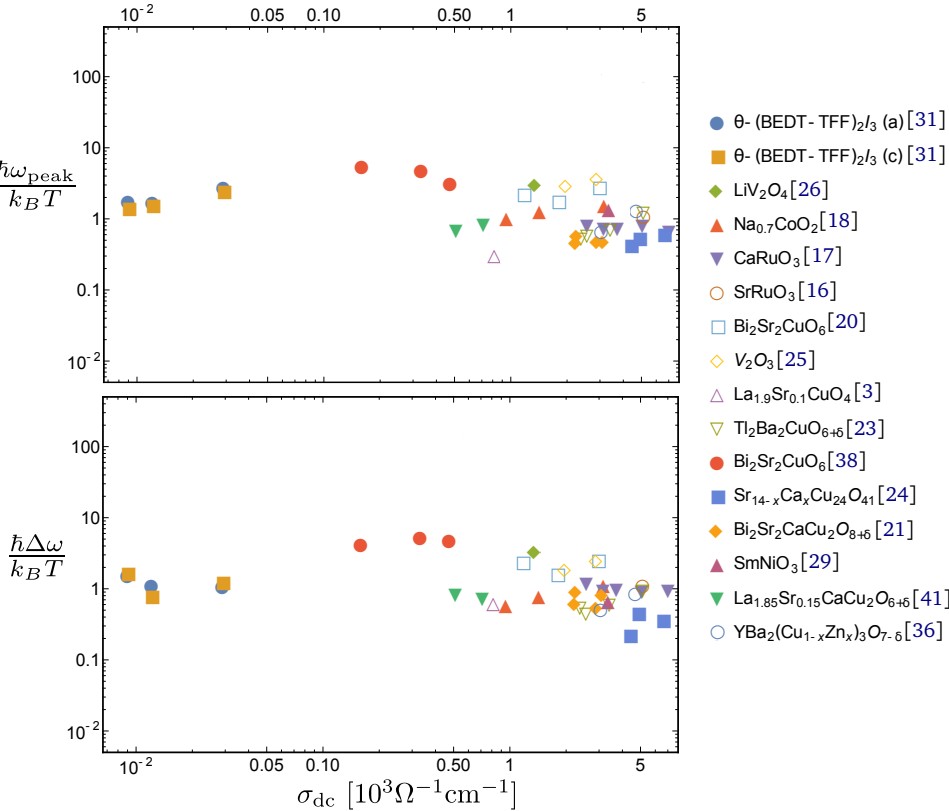

Figure 2: Location and width of the peak in $\sigma(\omega)$ versus dc conductivity bad metals.

diagram grows with increasing temperature, leading to 'quantum critical fans' within which $k_B T$ is the dominant energy scale. It has been argued that many bad metals occur in such quantum critical fans [44]. Furthermore, the phase diagrams of these systems are replete with various flavors of spin or charge density wave states at low temperatures. Quantum disordering of these translation symmetry breaking phases leads to quantum critical fans described by fluctuating density waves [45, 46]. The quantum critical rates of equation (3) therefore highlight the need to include phase relaxation in the hydrodynamic analysis, as we have done.

The time- and lengthscales of the fluctuating density waves can be estimated from (3). At a temperature of $T = 300$ K the timescales $\Omega$ and $\omega_o$ are both of order 25 fs. The spatial correlation can be estimated as $k_o \sim \omega_o/c$, where $c$ is the density wave shear sound or phason speed. With a characteristic cuprate electronic velocity of around $10^5$ m/s, at $T = 300$ K this leads to a short correlation length of order 25 Å. These numbers are broadly compatible with an extrapolation to higher temperatures of the observed length- and timescales of density waves in, for example, the cuprate pseudogap measured with x-rays [47] and ultrafast transient grating spectroscopy [48], respectively.[2]

In the cuprates the situation is complicated by the presence of the pseudogap and absence of measured translation symmetry breaking correlations near optimal doping, despite the observation of Fermi surface reconstruction across the critical point [49–51]. However, the quantum melting of density waves states can lead to candidate theories of both the pseudogap and strange metal phases, see e.g. [52–60]. Some of these works generalize the well

---

[2]The signal associated in [48] to the phason shows an overdamped decay in time, with no oscillations. This is compatible with optical conductivity data in the same compound (underdoped LSCO) over the same pseudogap temperatures [3, 22] where the Drude peak is still centered at $\omega = 0$, corresponding to $\omega_o < \frac{1}{2}\Omega$ in (20) (with $\Gamma = 0$).

established theory of classical, thermal melting transitions [61,62], leading to electronic liquid crystal phases [63].

The Planckian timescale $\tau \sim \hbar/(k_B T)$ has also been shown to underpin dc transport of charge [34] and heat [64] in several families of bad metals. In fact, given the frequencies (3) extracted from optical data, we can now use our hydrodynamic formula (2) to predict the dc conductivity and connect with these results. To do this, we consider a nearly Galilean invariant limit with weak momentum relaxation, so that $\sigma_o$ and $\Gamma$ can be neglected. We will want to keep $\omega_o$, however. The pinning frequency $\omega_o$ arises due to the fact that explicit translation symmetry breaking gaps the Goldstone mode. However, in the quantum critical regimes of interest, thermal and quantum symmetry restoring amplitude fluctuations are also important for $\omega_o$ (now viewed as a 'thermal mass'). Furthermore, in this nearly Galilean invariant limit we can write the charge density $\rho = -en$, with $e$ the fundamental charge and $n$ the electron number density, as well as the momentum susceptibility $\chi_{\pi\pi} = mn$, with $m$ the electronic mass. Then the formula (21) for the dc conductivity becomes

$$\sigma_{\text{dc}} = \frac{ne^2}{m} \frac{\Omega}{\omega_o^2} . \tag{4}$$

This is similar to the conventional Drude formula for the conductivity, except that in place of the quasiparticle or momentum relaxation time $\tau$, the ratio $\Omega/\omega_o^2$ has appeared.

The results in figure 2 show that $\Omega \sim \omega_o \sim k_B T/\hbar$. Equation (4) then predicts the dc resistivity

$$\rho_{\text{dc}} = \frac{1}{\sigma_{\text{dc}}} \sim \frac{m}{ne^2} \frac{k_B T}{\hbar} . \tag{5}$$

This is precisely the $T$-linear resistivity, including the correct prefactor, widely observed in bad metals [34]. That is to say, equation (4) relates a collection of independently measurable quantities, $n, m, \rho_{\text{dc}}, \Omega, \omega_o$, and this relation in fact holds experimentally – combining the analysis of [34] with our figure 2. The remarkable feature here is that a small momentum relaxation rate $\Gamma$ does not result in a small resistivity. This enables the Planckian timescale of the quantum critical fluctuations, $\Omega \sim \omega_o \sim k_B T/\hbar$, to feed directly into the resistivity, without being 'short circuited' by the long-lived momentum. Within a quantum critical fan, the behavior (5) can furthermore continue down to low temperatures where the resistivity will no longer be large.

We are therefore led to the following proposal that is consistent with the observed data for the optical and dc conductivities of many bad metals: as one moves out of a charge-ordered phase into the quantum critical fan, the optical conductivity takes the form (19) with $\omega_o \gtrsim \frac{1}{2}\Omega \sim k_B T/\hbar > \Gamma$ and with $\sigma_o$ small. The factor of one half is in order for the peak to remain off the $\omega = 0$ axis. The rates $\omega_o$ and $\Omega$ are due to quantum fluctuations of the amplitude and phase of the order parameter in the quantum critical regime. Strictly speaking these rates of order $k_B T/\hbar$ are at best on the boundary of a hydrodynamic regime describing momentum and Goldstone physics; in the quantum critical fan, $\sigma(\omega)$ should be computed from the effective long wavelength theory of the density wave quantum phase transition. While some computations of transport in a candidate quantum critical theory exist at $T = 0$ [65,66] and at $\omega = 0$ [67,68], the possibility of a peak in the optical conductivity at nonzero $\omega/T$ has not been considered. The peak location $\omega_o$ will be sensitive both to the pinning of the pseudo-Goldstone mode and to the thermal gap generated in the quantum critical theory. To repeat one of our main points again, an important virtue of this scenario is that the momentum relaxation rate $\Gamma$ can be small without the dc conductivity being large. This is the remnant of the charge density wave order in the fluctuating regime.

The way in which fluctuating density waves decouple dc transport from the lifetime of a collective mode can usefully be expressed as a failure of the intuition behind the extended

Drude model. In the extended Drude model, the conductivity is expressed as [69]

$$\sigma(\omega) = \frac{1}{4\pi} \frac{\omega_p^2}{1/\tau(\omega) - i\omega(1 + \lambda(\omega))} \,. \tag{6}$$

In particular, $1/\tau(\omega)$ is conceived of as a frequency-dependent scattering rate. The plasma frequency $\omega_p^2 = 8 \int_0^\infty \mathrm{Re}\,\sigma(\omega)d\omega$ is in general sensitive to the microscopic completion of the hydrodynamic $\sigma(\omega)$. The dc resistivity is determined by $\rho_{\mathrm{dc}} \sim 1/\tau(0) \sim T$, with the rate increasing with temperature as appropriate for a metal. In contrast, the fluctuating charge density wave optical conductivity in (1) typically leads to a $1/\tau(\omega)$ that decreases with increasing small frequency before eventually turning around and increasing at larger frequencies. For instance, in the simplified limit discussed above in which $\sigma_o \sim \Gamma \sim 0$:

$$\frac{1}{\tau(\omega)} = \frac{\omega_p^2}{4\pi} \mathrm{Re} \frac{1}{\sigma(\omega)} = \frac{\omega_p^2}{4\pi} \frac{\chi_{\pi\pi}}{\rho^2} \frac{\Omega \omega_o^2}{\omega^2 + \Omega^2} = \rho_{\mathrm{dc}} \frac{\omega_p^2}{4\pi} \frac{1}{1 + \omega^2/\Omega^2} \,. \tag{7}$$

The optical conductivity (1), with $\Gamma = \sigma_o = 0$, has been used in the second equality. This expression obeys $\omega/T$ scaling when $\Omega \sim T$, as in the cases shown in figure 2, but decreases rather than increases as $\omega$ is increased from $\omega = 0$. This dramatically illustrates the dichotomy between dc transport and dynamic timescales in these systems. Dips in the low frequency $1/\tau(\omega)$ should arise for all of the materials we have been discussing, whose optical conductivity takes the form sketched in figure 1. For example, they are visible in the extended Drude fits to the conductivity of Bi-2212 in [21] and of $SrRuO_3$ in [16]. The simplified expression (7) is monotonically decreasing with $\omega$. A raise at larger frequencies is seen when nonzero $\sigma_o$ is included. Of course, there will be a crossover to non-hydrodynamic behavior at frequencies $\hbar\omega \gg k_B T$.

Clearly, the picture of bad metals we have presented motivates searching for direct experimental signatures of fluctuating density wave order across the many different families of bad metals [70]. Static and fluctuating charge density waves have been directly imaged in the cuprate pseudogap via e.g. STM [71], NMR [72], x-ray scattering [47, 73–75] and ultrafast spectroscopy [48, 76] probes. Recent experiments have also revealed charge order in overdoped cuprates [77]. In some cases the peak in the optical conductivity of bad metals continues to exist in underdoped and overdoped samples [21]. This may allow a quantitative comparison with results from direct imaging techniques. The order detected using these techniques has so far not extended into the bad metal regions of the phase diagram. The techniques directly probe the dynamics of spatial modulations in the charge density, which is ultimately controlled by the dynamic structure factor of the charge density $S(\omega, k)$, see e.g. [70]. The singular contribution to the structure factor close to the ordering wavevector can be computed from hydrodynamics and comes from the pseudo-Goldstone mode (i.e. pinned phason). The singular contribution is therefore controlled by the same length- and timescales as found and discussed around (3) above. Crucially, however, the intensity of the peak in the structure factor is proportional to the amplitude of the density wave condensate. This amplitude is essentially 'pure fluctuation' in the quantum critical regimes we are interested in, and hence the intensity of the peaks is small and the signals detected in the above experiments disappear as the quantum critical fan is entered.

In contrast to $S(\omega, k)$, we see in equation (1) that the intensity of the peak in the optical conductivity is set by the Drude weight $\rho^2/\chi_{\pi\pi}$, independently of the magnitude of the condensate. The density wave stiffness instead appears in the optical conductivity through the timescale $\omega_o$ in (14). This means that even strongly fluctuating density wave order can leave a substantial imprint in time-resolved current dynamics. The role of fluctuating density waves for bad metals will therefore be further elucidated by a more detailed exploration of current dynamics. In particular, the presence of fluctuating density waves leads to concrete signatures in spatially resolved current dynamics [40], in the behavior of currents in a magnetic field [78], and in the nonlinear current response [15].

## Acknowledgements

We have benefitted greatly from discussions with Steve Kivelson and Jan Zaanen.

**Funding information**    This work is partially supported by a DOE Early Career Award (SAH), by the Knut and Alice Wallenberg Foundation (AK), by the Marie Curie International Outgoing Fellowship nr 624054 within the 7th European Community Framework Programme FP7/2007-2013 (BG) and by the Swiss National Science Foundation (LVD). The work of BG was partially performed at the Aspen Center for Physics, which is supported by National Science Foundation grant PHY-1066293.

## A  Hydrodynamic derivations

The basic ingredients of hydrodynamics are (i) approximate or exact conservation laws, (ii) constitutive relations for currents and (iii) 'Josephson' relations for approximate or exact Goldstone modes [4]. Hydrodynamics with spontaneously broken translational invariance is technically involved because rotational invariance is also typically broken and many terms appear in the equations. We shall only outline the ingredients going into the computation of $\sigma(\omega)$ here. More details can be found in [40]. For simplicity, we will focus here on the case of a smectic (uni-directional) charge density wave and consider current moving in the modulated direction. We take this to be the $x$ direction. Other cases are qualitatively similar [40]. The equilibrium physics of different patterns of symmetry breaking is described by the theory of elasticity, see e.g. [63], and dissipative hydrodynamics is built on top of that. The smectic case is anisotropic, so a bidirectional charge density wave will be necessary to have bad metallic behavior in all directions.

The hydrodynamic degrees of freedom that describe fluctuations about the equilibrium state, and that are relevant for the computation of $\sigma_{xx}(\omega)$, are: fluctuations in the charge density $\delta\rho$, fluctuations in the velocity $v_x$ and fluctuations of the phase of the charge density wave $\phi$. We will assume for simplicity that heat diffusion is decoupled from charge diffusion (i.e. that thermoelectric effects are small). It is often useful to work with local fluctuations of the chemical potential, that are related to charge density fluctuations as $\delta\rho = \chi\,\delta\mu$, where $\chi$ is the compressibility. Similarly the velocity fluctuations are related to momentum density fluctuations by $\delta\pi_x = \chi_{\pi\pi} v_x$. In Galilean-invariant theories the susceptibility $\chi_{\pi\pi} = nm$, with $n$ the density and $m$ the mass, but not in general.

We will first describe the hydrodynamics of charge density wave states. Later, we will also want to include the effects of phase disordering by proliferating dislocations.

The constitutive relation for the electric current $j$ expresses the current as a derivative expansion of the hydrodynamics modes. The terms that are relevant for $\sigma_{xx}(\omega)$ are

$$j_x = \rho\, v_x - \sigma_o \partial_x \delta\mu + \gamma_o \kappa\, \partial_x^2 \phi + \cdots . \tag{8}$$

Here $\rho$ is the equilibrium charge density, $\kappa$ is the stiffness of the charge order and $\sigma_o$ and $\gamma_o$ are dissipative coefficients ($\sigma_o$ must be positive, we will discuss constraints on $\gamma_o$ below). In a Galilean-invariant theory, $j_x \propto \delta\pi_x$ holds as an operator equation (see e.g. [4]) and hence $\sigma_o = \gamma_o = 0$ in that case. We will be interested for the moment in non-Galilean invariant theories. For instance, theories describing the interactions of fermions with quantum critical bosons are typically not Galilean invariant [45]. The physical content of (8) is transparent. Current is caused by motion, by charge gradients and by a compression gradient in the charge order. In this sense the last term $\gamma_o$ controls a kind of dissipative piezoelectric effect. The association of currents to gradients typically results in diffusive processes, as we shall see.

The relevant terms in the 'Josephson relation' for $\phi$ are

$$\dot{\phi} = \nu_x + \gamma_o \partial_x \delta\mu + \cdots . \tag{9}$$

The first, nondissipative, term in this expression mirrors the case of superfluids, where the Josephson relation is $\dot{\phi} = -\mu$. In that case $\mu$ is conjugate to charge density $\rho$ that generates the spontaneously broken symmetry. In our case $\nu_x$ is conjugate to the momentum density $\pi_x$ that generates the spontaneously broken translation symmetry. The appearance of $\gamma_o$ in (9) is dictated by Onsager relations.

With no explicit breaking of translation invariance, equations (8) and (9) are to be combined with the equations for conservation of charge and momentum to obtain a closed set of hydrodynamic equations. Weak, explicit breaking of translation invariance has two effects. Firstly, the phase $\phi$ becomes a gapped pseudo-Goldstone boson. This means that the free energy of the phase changes as

$$f = -\frac{\kappa}{2}\phi\,\partial_x^2\phi \quad \rightarrow \quad f = \frac{\kappa}{2}\phi\left(-\partial_x^2 + k_o^2\right)\phi . \tag{10}$$

The inverse spatial correlation length $k_o$ then appears in the constitutive relation (8), where one must replace $\partial_x^2\phi \to (\partial_x^2 - k_o^2)\phi$. Secondly, conservation of momentum is weakly broken so that

$$\dot{\pi}_x + \partial_x \tau_{xx} = -\Gamma \pi_x - \kappa k_o^2 \phi + \cdots . \tag{11}$$

Here $\Gamma$ is the momentum relaxation rate, the appearance of the last term is again required by Onsager relations and the relevant terms in the stress tensor are

$$\tau_{xx} = p - \kappa \partial_x \phi + \cdots . \tag{12}$$

Here $p$ is the pressure. Finally, we will see later that positivity of entropy production with momentum relaxation but no phase fluctuations (mobile dislocations) requires $\gamma_o = 0$. We set $\gamma_o$ to zero in the following few formulae, it will reappear later when we discuss phase disordered charge density waves.

The optical conductivity $\sigma(\omega) \equiv \sigma_{xx}(\omega)$ is obtained from the above equations following the method of Kadanoff and Martin [5]. The answer is

$$\sigma(\omega) = \sigma_o + \frac{\rho^2}{\chi_{\pi\pi}} \frac{-i\omega}{-i\omega(\Gamma - i\omega) + \omega_o^2} . \tag{13}$$

Here we introduced the frequency

$$\omega_o^2 \equiv \frac{\kappa k_o^2}{\chi_{\pi\pi}} . \tag{14}$$

In equation (13) we have dropped terms that are subleading in a hydrodynamic expansion [40].

The poles in the denominator of (13) occur at the complex frequencies

$$\omega_\star = \pm\sqrt{\omega_o^2 - \tfrac{1}{4}\Gamma^2} - i\tfrac{1}{2}\Gamma . \tag{15}$$

If $\omega_o^2$ is sufficiently large compared to $\Gamma$ then the pole has a nonzero real part and the peak in the optical conductivity moves off the $\omega = 0$ axis. Note that (15) depends only on the way translation invariance is broken and on the stiffness of the charge density wave. In contrast, the dc conductivity is

$$\sigma_{\text{dc}} = \sigma_o . \tag{16}$$

Recall that $\gamma_o$ has been set to zero. The conductivity (16) is independent of the manner and strength of translation symmetry breaking and also on the stiffness. The dc conductivity and

the location of the peak in the optical conductivity are described by completely orthogonal physics.

The dc conductivity (16) admits an interpretation in terms of universal diffusive transport. The hydrodynamic modes following from the equations above are easily computed [40]. At the longest frequencies and wavelengths, $\{\omega, k^2\} \ll \{\Gamma, k_o^2\}$, one finds that the only surviving hydrodynamic modes are diffusion of charge and entropy. As we are neglecting thermoelectric effects, the charge diffusivity $D$ is found to satisfy $\sigma_{\text{dc}} = \chi D$. The dc conductivity is controlled by diffusion. This is similar to the strong momentum relaxation scenario discussed in [6]. The new feature is that even with weak momentum relaxation, the diffusivities can be small.

The effect of phase-disordering mobile dislocations is to modify the Josephson relation (9). Because the true hydrodynamic variable is the gradient of the phase, one should add the phase relaxation rate $\Omega$ into the gradient of the Josephson relation, which now becomes

$$(\partial_t + \Omega)\partial_x \phi = \partial_x v_x + \gamma_o \partial_x^2 \delta\mu + \cdots . \tag{17}$$

In the presence of $\Omega$, the dissipative coefficient $\gamma_o$ can be nonzero but is constrained by positivity of entropy production to satisfy

$$\gamma_o^2 \le \frac{\sigma_o \Omega}{\chi_{\pi\pi} \omega_o^2} . \tag{18}$$

In fact $\gamma_o$ is subject to a stronger bound [40], but the above bound is sufficient for our purposes. In particular, as $\Omega \to 0$ then $\gamma_o$ is forced to zero, as we noted previously.

With the modified Josephson relation (17), the conductivity is computed as outlined above. One finds that (13) is generalized to

$$\sigma(\omega) = \sigma_o + \frac{(\Omega - i\omega)\rho^2/\chi_{\pi\pi} + 2\rho\gamma_o\omega_o^2}{(\Omega - i\omega)(\Gamma - i\omega) + \omega_o^2} . \tag{19}$$

The bound (18) ensures that the real, dissipative part of $\sigma(\omega)$ is positive. In fact, the bound (18) means that the $\gamma_o$ term in (19) leads to no qualitatively new physical effects in (19) and hence for clarity we have set it to $\gamma_o = 0$ in the main text. The position and width of the peak (15) now depends on both $\Gamma$ and $\Omega$

$$\omega_\star = \pm\sqrt{\omega_o^2 - \tfrac{1}{4}(\Gamma - \Omega)^2} - i\tfrac{1}{2}(\Gamma + \Omega), \tag{20}$$

and the dc conductivity is now

$$\sigma_{\text{dc}} = \sigma_o + \frac{\Omega\rho^2/\chi_{\pi\pi} + 2\rho\gamma_o\omega_o^2}{\Omega\Gamma + \omega_o^2} . \tag{21}$$

It is clear that when $\Omega = 0$, and hence $\gamma_o = 0$, one recovers the previous result (16).

## B   Temperature dependence of the peak location

Figure 3 shows the peak frequency as a function of temperature for the same bad metals as were shown in figure 2 in the main text. It is again a log-log plot. The data points are the same, but this plot shows the actual values of the frequencies and temperatures for the data. Furthermore, in this plot we have included the two manganate compounds. These do not quite fit into the paradigm outlined in the main text. The peak frequencies have a significantly stronger than linear dependence on temperature and $\omega_{\text{peak}}/T$ is significantly larger than the data points shown in figure 2 in the main text. Because $\hbar\omega_{\text{peak}} \gg k_B T$, these peaks cannot be captured by hydrodynamics.

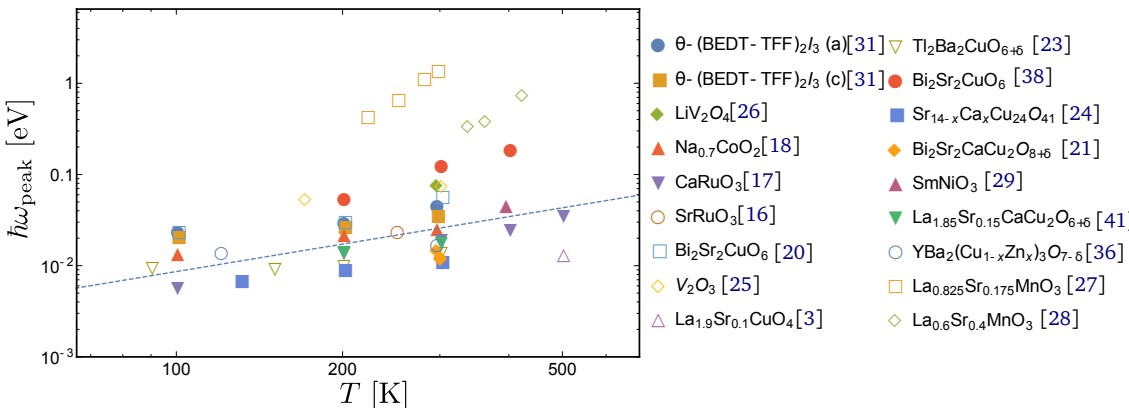

Figure 3: Location of the peak in $\sigma(\omega)$ versus temperature in bad metals. The dashed line shows $\hbar\omega = k_B T$.

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
