# Peer review of "Bad Metals from Fluctuating Density Waves"

_SciPost Physics, doi:SciPost Phys. 3, 025 (2017)_

## Round 4 · Referee Report · Anonymous (Referee 1) · 2017-7-22

Strengths

This work represents a fresh eye on an old subject. What to expect for the conductivity of a fluid subjected to relatively weak breaking of Galilean invariance, that is however characterized by strong commensurate thermally fluctuating density-density correlations? This was already hotly pursued in the seventies (Fukuyama-Lee-Rice-Anderson) and it revived in the 1990's as related to fluctuating stripes. The hard work was actually done by the present authors in ref. [40]: in a characteristic style, completely resting on hydrodynamical principles, they derived eq. 1 for the optical conductivity. The surprise is that for a growing pinning energy a peak develops at finite frequency in the optical conductivity (Fig. 1). In hindsight this makes much sense but I was caught by surprise when I saw it for the first time, and it seems that in this regard I was not the only one. An evolution of the kind shown in Fig 1 as function of increasing temperature is measured in an variety of metallic oxides and organics (see Fig. 2) and the authors make the case that these electron fluids could be generically characterised by such strong crystalline correlations. This is a highly provocative idea; I am quite sceptical but its merit is that it may generate attempts to test it experimentally. Given the recent advances in experimental techniques it should become possible to try to falsify in a near future. When it turns out to be correct it would completely change the basic outlook on bad metal physics. In summary, this type of thinking out of the box is highly desired in this branch of physics.

Weaknesses

There are many weaknesses: I am actually very sceptical. But I can't prove the authors to be wrong and it is just bad taste to use such guts feelings to censor papers. I repeat, this will likely provoke new experimental activity and I would cherish it when it turns out that this paper is on a right track.
Some of the reasons: (1) I wonder to what extent these authors realise the established principe that "transport is the first thing that one measures and the last thing that is explained." The issue is that there is not much information in (optical) conductivities while it may be sensitive to many aspects of the complex physics in oxides etc. As a veteran of condensed matter phenomenology I would be myself most hesitant to build a case as the present solely resting on transport information. (2) At the least one should attempt to arrive at high quality fits of the data itself with the Eq. 1 in the text. Dealing with real data it is just a bit very hand waiving to not get beyond the observation that "there is some peak in the data". (3) Also in this regard, transport would be the last alley to look for firm evidence for strong charge order correlations in the fluid. This can be measured directly using RIXS and EELS. At least in the cuprates, although clear signals are now picked up in the under doped- and overdoped regime the latest data indicate that up to room temperature optimally doped cuprates (showing the linear-in-T resistivity) are devoid of any correlations of the kind. To my eyeballs, the theory of the authors gets the transport just quite wrong in the regime where charge correlations have been established (underdoped, also the CDW organics). (4) A very interesting result by itself is the scaling of the peak energies and width, determined from experimental data (Fig. 3). The intriguing finding is that both increase linearly in temperature, being even set quantitatively by the "Planckian dissipation" principle. The authors express the hope that this will be eventually explained in terms of quantum critical states involving charge order, whatever. To the best of my understanding this is impossible for very simple reasons. The "first law" governing pinning energies insists that these are proportional to the strength of the charge-order correlations. This is canonical; in the conventional CDW cases one finds that the pinning energy goes to zero when the (mean-field) order parameter disappears at Tc; typically one may find a transition from the pinned (commensurate) phase to a truly incommensurate state when temperature is raised. In order to see the increase of \omega_0 with temperature it has to be that the charge correlations are {\em growing} when temperature increases. Regardless whether quantum fluctuations are in one or the other way in the game, this defeats general statistical physics principle: charge correlations represent order and when temperature increases entropy will invariably win when temperature gets high enough. (5) Specifically for the cuprates, it is just a myth that their secrets reside in bad metal behaviour. It is correct that above \sim 400 K the strange metals are nominally bad metals but at lower temperature they turn into good metals and at very low temperature they appear to become {\em excellent} metals (absence of residual resistivity, microns mean free path in quantum oscillations). What needs to be explained is that there is no sign of cross-overs between these different regimes: the resistivity is perfectly linear. It is odd to address this in terms of theories that are geared up to explain only {\em high} resistivities.

Report

See the "strengths" and "weaknesses". Whatever happens, at the least Fig.2 will be remembered: even ignoring the interpretations put forward by the authors this is a very intriguing, purely phenomenological observation that has not been noticed before.

Requested changes

I leave it to the authors to use some of my observations to improve the text. When they have the energy they may attempt to actually use eq. 1 to fit some real data to see whether there is more to the story than just a peak.

---

## Round 4 · Referee Report · Anonymous (Referee 2) · 2017-8-21

Strengths

The manuscript makes several interesting points about applying the hydrodynamics of phase fluctuating density waves to describe the conductivity of bad metals.
By extracting the location of the displaced peak and its width from data for a variety of bad metals, the paper identifies a simple dependence of both quantities on the Planckian timescale, a characteristic signature of quantum critical systems.
This elegant result provides evidence for the importance of including phase relaxation in the hydrodynamics analysis, as the authors stress. They also do a good job highlighting the crucial roles played by different scales in the system, and in particular the interplay between momentum dissipation and phase relaxation.
Finally, the paper makes concrete predictions for experimental signatures of fluctuating density wave order and clearly motivates searching for them in the bad metal region of the phase diagram of many materials.

Weaknesses

I have no objection to the physics, but I think the results would be easier to appreciate and understand if the paper was structured slightly differently, with more material from the supplemental sections included in the main text, as a minimal change.
As is, some of the discussion comes across as being rather scattered, and it is hard for the reader to easily see a coherent picture and appreciate the significance of all the points made by the authors.
The discussion of the way in which the location and width of the peaks were extracted, as well as the extended Drude formalism, could easily be moved to the main text. This is a minimal change which would improve the manuscript.

Report

This is a very interesting paper which discusses the temperature dependence of the optical conductivity for several families of bad metals, focusing on understanding the shift of the Drude peak away from zero frequency with increasing temperature. The goal is to identify collective low energy dynamics and provide a unified description of transport in these systems. The analysis relies on the optical conductivity results obtained by the authors in their earlier paper [40], in which they examined the hydrodynamics of charge density waves, taking into account the effects of broken translational invariance as well as phase relaxation. Inspecting the available conductivity data for a variety of bad metals, the authors extract the location of the displaced Drude peak and its width (see Fig. 2 and Eq. 3). The dependence of both quantities on the Planckian timescale, apparent from Eq.3, is characteristic of a quantum critical system. One of the crucial points of the analysis is to emphasize that the hydrodynamics of phase fluctuating (charge) density waves can describe the features seen in the transport properties of bad metals, and to stress in particular the need to include phase relaxation. The authors also identify concrete experimental signatures of fluctuating density wave order, such for example peaks in the frequency dependence of the scattering rate. The paper makes several valuable points about the effectiveness of the hydrodynamic description of density wave order - and the role of phase relaxation in addition to momentum dissipation - to understand the behavior of bad metals.

Requested changes

To improve the clarity of the discussion, I suggest that the authors incorporate the contents of Sections B and C into the main text.
I leave it up to them to do so.

---

## Round 5 · Author Response

We thank both of the referees for their encouraging positive comments and constructive criticism.

In response to referee 2 we have moved all of appendix C and most of appendix B into the main text. We agree that this improves readability. We kept the plot in appendix B as an appendix because we feel this plot is not essential for a first reading of the paper. We kept the derivations in appendix A as an appendix because we believe that keeping the main text relatively decluttered from equations will make it more accessible to experimentalists who may wish to use our formulae for fitting data.

We would like to make the following comments in reply to the very reasonable questions raised by referee 1. Here we have only made two minor additions to the text, relating to the points (3) and (5) below. The remaining points are all explicitly addressed already to varying extents in the main text. We hope that having these replies available online together with the referee’s comments will be sufficiently useful to readers for whom the same questions may arise.

(1) The first concern is that transport is an insufficient foundation to build a theory of the complex physics at work (despite the fact that regimes we are studying — bad metals — are precisely defined by their transport behavior). In fact, as the referee surely knows well and as we noted in the text, especially the final two paragraphs, fluctuating charge density waves have been argued to be important across the phase diagram of e.g. cuprates for reasons that have nothing to do with transport. See e.g. figure 1 of the review by Kilveson et al. There have been explicit recent observations of CDWs in overdoped, as well as underdoped cuprates. Therefore the invocation of fluctuating CDWs as underpinning the bad metal regime is not completely without context. Further, various of the other materials in our list show very compelling evidence for SDW order close to the bad metal regime. While our theory as developed does not directly apply to SDWs, it is the case that SDWs also involve spontaneous breaking of translation invariance and so we expect that some of the same physics will be at work. But, indeed, the main point of our paper is that there is a forceful logic for translational order that operates purely in the realm of sigma(w) — it is nontrivial to displace the Drude peak!

(2) The second concern is that one should be able to fit the data in detail using our functional form, rather than just extract the peak locations and widths. We have preferred to go for a “low-tech” extraction of the peak characteristics in order to minimize human interference with the interpretation of the data. Careful fitting of the peak will likely involve subtracting out other non-hydrodynamic components in sigma(w), and this can be an involved task. At the time of writing this reply we are aware of at least one experimental group that is successfully using our form of sigma(w) to fit their data and we hope that there will be more in the future.

(3) The third concern is that (a) one should look for CDWs in X-rays etc. before transport and there is no evidence of an X-ray signal in bad metal regimes and (b) in regimes where there is an X-ray signal of CDWs our expressions do not match the transport. Regarding the first of these concerns, in fact, one of the points we emphasized in the text (in the final paragraph) is that when the magnitude of the CDW condensate becomes small, sigma(w) becomes a more powerful diagnostic than X-rays! This is because the weight of the displaced Drude peak in sigma(w) is set by the Drude weight and, unlike the peaks in X-ray data, is not proportional to the condensate. This allows the peak to be pronounced even with a small condensate. We have suggested in the text that looking at sigma(w,k) experimentally would amount to the best of both worlds — a stronger peak than in X-rays together with an explicit connection to spatial dependence. Regarding the second concern, in this paper we have preferred to focus on bad metal regimes only as these are the most similar between different materials and hence are good candidates for “universal” hydrodynamic reasoning. However, in our longer and more technical paper (ref 40) we have noted that aspects of underdoped cuprate transport, such as resistivity upturns, may possibly admit an explanation in terms of density wave dynamics. Furthermore, some optical data in the underdoped regime of cuprates (eg. our ref [21]) show peaks that continue into the bad metal regime. These peaks will fit within our theory, so long as the frequencies are low enough to admit a hydrodynamic interpretation. Of course this regime is not quantum critical and so the temperature dependence of the various hydrodynamic parameters will be different. There are many non-hydrodynamic peaks seen in the optical conductivity of underdoped samples, but because these are at non-hydrodynamic energy scales our approach cannot tie them CDW physics as opposed to other pseudogap related phenomena.

—> We added the sentence, in the second to last paragraph, “In some cases the peak in the optical conductivity of bad metals continues to exist in underdoped and overdoped samples [21]. This may allow a quantitative comparison with results from direct imaging techniques.”

(4) The fourth concern is that the pinning frequency is proportional to the condensate strength and this should not increase with temperature. This is a fair point, and we agree that at present we know of no microscopic theory in which the pinning frequency increases with temperature. However, as we note in the text (in the paragraph above equation 6), if these peaks are indeed within quantum critical fans, then pinning due to disorder is not the only dynamics at work. There will also be temperature-dependent “thermal masses” generated in the quantum critical theory. A true microscopic computation will need to take into account the interplay between this quantum critical effect and pinning. I.e. the question is whether the quantum critical Planckian mass scale can “drag” the Drude peak out to it. This will require the inclusion of disorder effects at the quantum critical point for e.g. charge density waves in a metal. One of us is currently investigating this question -- it is a well-defined technical question -- but it is beyond the scope of the present paper.

(5) The fifth concern is that bad metallicity occurs only at high temperatures wheres, at least in cuprates, the dc transport behavior is remarkably continuous all the way down to low temperatures. This is a fair point and indeed the persistence of the T-linear resistivity from the lowest to highest temperatures in bad metals is perhaps *the* central mystery of transport in the cuprates. Even if our theory of bad metals is experimentally corroborated, we will not necessarily have solved that more general problem. That said, in principle our expressions (4) and (5) for the conductivity can hold down to arbitrarily low temperatures in the quantum critical fan. While fluctuating CDWs offer an avenue to bad metallicity, by displacing the Drude peak, they are also compatible, in principle, with good metallicity if quantum fluctuations are strong enough.

—> We added the following sentence at the end of the paragraph containing equation (5): “Within a quantum critical fan, the behavior (5) can furthermore continue down to low temperatures where the resistivity will no longer be large.”

---

## Round 5 · List of Changes

(a) Appendix C and most of Appendix B moved to main text.

(b) We added the sentence, in the second to last paragraph, “In some cases the peak in the optical conductivity of bad metals continues to exist in underdoped and overdoped samples [21]. This may allow a quantitative comparison with results from direct imaging techniques.”

(c) We added the following sentence at the end of the paragraph containing equation (5): “Within a quantum critical fan, the behavior (5) can furthermore continue down to low temperatures where the resistivity will no longer be large.”

---

## Editorial Decision

published